# Prevalence of enteric pathogens, intestinal parasites and resistance profile of bacterial isolates among HIV infected and non-infected diarrheic patients in Dessie Town, Northeast Ethiopia

**Assefa Belay[1], Melaku Ashagrie🄳[2], Berhanu Seyoum[3], Mekuanent Alemu[1], Aster Tsegaye🄳[4] ***

**1** Department of Medical Laboratory Sciences, Dessie Health Science College, Dessie, Ethiopia,
**2** Department of Medical Laboratory Sciences, College of Medicine and Health Science, Wollo University, Dessie, Ethiopia, **3** Armauer Hansen Research Institute, Addis Ababa, Ethiopia, **4** Department of Medical Laboratory Sciences, College of Health Sciences, Addis Ababa University, Addis Ababa, Ethiopia

* tsegayeaster@yahoo.com, aster.tsegaye@aau.edu.et

**Data Availability Statement:** All relevant data are within the manuscript.

**Funding:** The author(s) received no specific funding for this work.

## Abstract

### Background

Enteric pathogens like *Salmonella* and *Shigella* species as well as intestinal parasites (IPs) are among the main causative agents of diarrhea in people with human immunodeficiency virus (HIV)/acquired immune deficiency syndrome (AIDS), particularly in low income countries like Ethiopia. Antimicrobial resistance against commonly prescribed drugs has become a major global threat. This study, therefore, aimed at determining the magnitude of *Salmonella*, *Shigella* and IPs infections, their predicting factors, and antimicrobial susceptibility pattern among HIV infected and non-infected diarrheic patients in Dessie town, Northeast Ethiopia.

### Methods

A cross sectional study was conducted at three health facilities in Northeast Ethiopia between January 2018 and March 2018. Data on socio-demographic and associated risk factors were collected using structured questionnaire from 354 HIV infected and non-infected diarrheic outpatients. Fresh stool specimen was processed according to standard operating procedures. Data were entered and analyzed using SPSS version 22. Descriptive statistics was used to determine frequency, Bivariate and multivariate logistic regression analyses were performed to identify predicting factors associated with the outcome variable. P-value <0.05 were used to declare statistical significance.

### Results

Among 354 diarrheic patients, 112 were HIV infected and 242 were HIV non-infected. The overall prevalence of intestinal parasite and bacterial infection among HIV infected *versus*

**Competing interests:** The authors have declared that no competing interests exist.

**Abbreviations:** AIDS, Acquired immune deficiency syndrome; AML, Amoxicillin; AMP, Ampicillin; ART, Antiretroviral therapy; AST, Antimicrobial Susceptibility Testing; CAF, Chloramphenicol; CD4, Cluster of Differentiation 4; CDC, Center for Disease Control and prevention; CIP, Ciprofloxacin; CLSI, Clinical Laboratory Standards Institute; DHRL, Dessie Health Research Laboratory; HIV, Human immunodeficiency virus; KIA, Kligler Iron Agar; MDR, Multi-drug resistance; NTS, Non typhoid Salmonella; SPSS, Statistical package for social science; SXT, Sulfamethoxazole Trimethoprim; WHO, World health organization; XLD, Xylose Lysine Deoxycholate agar.

non-infected, respectively, was 26 (23.2%) and 8 (7.1%) *versus* 50 (20.7) and 16 (6.6%). *Salmonella* was the highest in both groups, 6 (5.4%) vs 11 (4.5%). Most prevalent parasite was *C. parvum*, 9 (8%) among HIV+ while *E. histolytica/dispar* 39 (16.1%) among HIV-. Having bloody plus mucoid diarrhea, not utilizing latrine and drinking river or spring water were factors significantly associated with bacterial infection. Whereas, being illiterate or having primary level education, diarrhea lasting for 6–10 days, CD4 level between 200–500 cells/μl, not washing hand with soap showed significant association with IPs. The bacterial isolates were 100% susceptible to Ceftriaxone and 95.4% to Ciprofloxacin, while 100% resistant to Ampicillin and Amoxicillin. MDR was observed among 19 (79.2%) isolates.

## Conclusion

Preventing and controlling infection by enteric pathogens as well as IPs require strengthening intervention measures. The 100% resistance of isolates to commonly prescribed antibiotics calls for expanding antimicrobial susceptibility testing so as to select appropriate antimicrobial agent and prevent emergence of drug resistant bacteria.

## Background

Diarrhea, as per the definition of World Health Organization (WHO) is the passage of three or more loose or liquid stools per day, or more frequently than is normal for an individual [1]. Diarrhea is still a global problem particularly in areas where access to safe water is limited and poor hygiene and sanitation are commonly practiced [2–5]. For example, in 2016, it was the eighth leading cause of death for more than 1.6 million people at global level. More than a quarter (26.93%) of the deaths occurred among under-fives. Moreover, about 90% of diarrheal deaths occurred in south Asia and sub-Saharan Africa [2].

Etiologies of diarrhea could be bacterial, viral, or parasitic. People living with human immunodeficiency virus (HIV) as well as children who are malnourished or have impaired immunity are most at risk of life-threatening diarrhea [1, 6–8]. HIV related immunosuppression, favors the occurrence of multiple opportunistic infections (OIs) which are responsible for a high morbidity and mortality. Among these OIs, intestinal parasites such as *Coccidia* (*Cryptosporidium parvum*, *Isospora belli*, *Cyclospora* sp) and amoebae (*Entamoeba histolytica/ dispar*) are the main cause of severe chronic diarrhea [9, 10] while *Salmonella* and *Shigella* are potential enteric bacterial agents regardless of immunocompetency status of an individual [8].

With an almost 90% occurrence of HIV/AIDS in adults and children in low income countries, persistent diarrhea is associated with an 11-fold increase in mortality compared to uninfected persons [7, 8, 10].

Globally, *Salmonella* is one of the most frequently isolated foodborne pathogens causing diarrhea. There are an estimated 11–21 million cases of typhoid fever and approximately 128 000–161 000 deaths annually. Sub-Saharan Africa is among those harboring the majority of the cases. An estimated 2.1–6.5 million cases of invasive *non typhoidal Salmonella* (iNTS) disease occur annually, with the highest incidence in Africa [4, 11]. The case fatality rate is high in those with HIV infection. Salmonellosis has been estimated to be nearly 20 times as common and 5 times more bacteremia in patients with HIV/AIDS than in those without the disease [12]. On the other hand, *Shigella* species are among the bacterial pathogens most frequently isolated from patients with diarrhea. Globally, *Shigella* is estimated to cause 80–165

million cases of disease and 600,000 deaths annually [13]. The majority of the burden occurs in developing countries including Ethiopia, primarily among children and adults [14, 15].

In general, immunosuppressed individuals are more likely to be susceptible to *Shigella* and *Salmonella* infections than healthy individuals. In HIV-infected adults the rates of Gram-negative bacteria enteric infections are at least 10-fold higher than in the general population. HIV infection increases the risk of Salmonella bacteremia by about 20- to 100-fold and mortality as high as 7-fold compared to those who are not HIV infected [16].

Intestinal parasitic infections are also common among HIV patients globally and have been known to cause severe and life-threatening diarrhea [17]. Sub-Saharan Africa harbors remarkably high burden particularly among HIV positive patients [9].

Antimicrobial resistance is a global threat. Even more alarming to developing countries like Ethiopia is the development of resistance to commonly prescribed drugs leaving physicians treating patients infected with enteric pathogens like *Salmonella* and *Shigella* species with little option. Besides, access to microbiology laboratories for culture and antimicrobial susceptibility testing are limited and hence empiric treatment of microbial infections is very common. Over the past few decades *Salmonella* and *Shigella* species have become progressively resistant to most of the first-line drugs used and the prevalence of multi drug resistant strains is an important concern of treatment [18–20]. The current treatment recommendation for Salmonelosis and shigelosis by WHO [21, 22] which is also adopted by Ethiopia as well [23] is ciprofloxacin for both infections or azithromycine for salmonellosis (where there is no ciprofloxacin). Continuous monitoring of the magnitude of these infections as well as antimicrobial resistance patterns is hampered by diagnostic challenges due to lack of adequate facilities. As a result, frequent and consistent evaluation and study of the prevalence, etiologic agents, and predisposing factors of enteric fever is necessary in developing countries like Ethiopia in order to reduce its devastating effects.

This study, thus, aimed at assessing the intestinal parasitic profile as well as prevalence and antimicrobial susceptibility patterns of *Salmonella* and *Shigella* species in Dessie city, North eastern Ethiopia, where there is scarcity of such data. The study also determined the predicting factors of infections with IPs and the two enteric pathogens.

## Materials and methods

### Study design, area and period

A health facility based cross-sectional study was conducted from January–March 2018 at health facilities in Dessie, Northeastern Ethiopia including Dessie Referral hospital, and two randomly selected health centers, namely, Dessie health center and Banbua wuha health center. Dessie is located 401kms north of the capital city, Addis Ababa. The city is one of the fastest growing urban areas in the country and covers an area of 15.08 square kilometers. The city has one referral hospital, one general hospital (Boru Meda), six health centers (Dessie, Buanbua Wuha, Segno Gebeya, Tita, Kurkur, Meytero), four private hospitals and several private clinics. Dessie Referral Hospital is one of the largest hospitals in the region which is providing diagnosis and treatment service for the community.

### Eligibility criteria

HIV infected and non-infected patients of both sexes and age greater than or equal to 15 years with diarrhea, who were willing to participate in the study were included. Those who were critically ill or patients on antibiotic or anthelminthic treatments within the previous 14 days were excluded from the study.

## Data collection and laboratory processing

Structured questionnaire was used to obtain information related to socio-demographic, clinical and risk factors data from 354 HIV infected and non-infected diarrheic out patients attending the medical and ART clinics of selected health facilities conveniently. Freshly passed stool specimen was collected using pre-labeled (date, time, identification code, age), leak proof, wide mouth, sterile, screw-capped plastic container (FL Medical, Italy) and clean wooden applicator stick after appropriate instructions were given. The collected specimens were then stored in cold box and transported to Dessie Health Research Regional Laboratory and processed within two hours.

## Stool examination of intestinal parasites

On a microscope slide, about 1–2 mg of stool was emulsified in a drop of normal saline (0.85% Na Cl) on the left hand side of the slide, and in Lugol's iodine on the right side of the slide. A cover-slip was then placed on each side, and the slides were scanned under 10× and 40× objective lenses of a light microscope, as required. Saline direct smear is used mainly for detection of motility of intestinal protozoan trophozoites, which are seen in liquid or semi-liquid specimens. Iodine direct smear shows the characteristic features of the diagnostic stages in more details and formol ether sedimentation concentration technique was used for detection of cysts, ova and larvae; and additionally modified Ziehl-Neelsen technique was used for identification of *Cryptosporidium* and *cyclospora* oocysts.

## Bacterial cultivation and identification

Freshly passed stool samples were placed using calibrated wire loop (0.001ml) in to Selenite F enrichment broth (Oxoid ltd, UK). After cultures were incubated overnight under aerobic condition at 37˚C for 24 hours, colonies were sub cultured onto Xylose Lysine Deoxycholate (XLD) agar (Oxoid ltd, UK), *Salmonella* and *Shigella* (SS) agar (Hi media) and then incubated at 37˚C for 18–24 hours. The growth of *Salmonella* and *Shigella* species was detected by their characteristic appearance on XLD agar (*Shigella*: red colonies, *Salmonella* red with a black center). Assessment was also made on SS agar (colorless colony) for *Shigella* and (colorless colony with black center) for *Salmonella*. The suspected colonies were further tested through a series of biochemical tests to identify *Shigella* and *Salmonella* species. The isolates were then characterized based on standard biochemical tests including Kligler iron agar (KIA), indole, urease, citrate utilization test, motility test, lysine decarboxylase (LDC) test [24]. Furthermore, all bacterial isolates were identified using standard clinical laboratory methods [24].

## Antimicrobial susceptibility testing

Antimicrobial susceptibility testing was performed using the standard Kirby-Bauer disk diffusion method recommended by Clinical and Laboratory Standards Institute (CLSI) [25]. Pure culture colonies of 24-hour growth were suspended in a tube with 4ml of physiological saline to get bacterial inoculums equivalent to 0.5 McFarland turbidity standards. Sterile cotton swab was dipped, rotated across the wall of the tube to avoid excess fluid and was evenly inoculated on Muller-Hinton agar (Conda ltd, USA) and then the antibiotic discs were placed on MHA plates. The following antimicrobials were used with their respective concentration: Amoxicillin (AML, 25μg), Ampicillin (AMP, 10μg), Tetracycline (TTC 30-μg), Trimethoprim-sulfamethoxazole (SXT, 25μg), Chloramphenicol (CAF, 30-μg), Naldixic acid (NAL, 30-μg) Ciprofloxacin (CIP,5-μg) and Ceftriaxone (CRO, 30μg). These antimicrobial drug disks are selected based on CLSI guide and also by considering the availability and

frequent prescriptions of these drugs for the treatment of *Salmonella* and *Shigella* in the study area. All antibiotic discs were from Oxoid, Ltd, UK. The plates were then incubated at 37˚C for 24 hours. Diameters of the zone of inhibition around the discs were measured using a digital caliper. The interpretation of the results of the antimicrobial susceptibility tests was based on the standardized table supplied by CLSI [25] criteria as sensitive, intermediate and resistant.

## Quality assurance

To generate quality and reliable data, all quality control checks were done before, during and after data collection. All the questions in structured questionnaire were prepared in a clear and precise way and translated into local language (Amharic). Data collectors were oriented how to collect data; the entire questionnaire was checked for completeness, during and after data collection by the principal investigator. Moreover, all laboratory assays were done by maintaining the quality control procedures. The raw data (the laboratory, clinical and demographic data) were checked for completeness and representativeness prior entry to the database. Standard Operating Procedures (SOPs) were strictly followed verifying that media meet expiration date and quality control parameters per CLSI guideline. Visual inspections of cracks in media or plates contamination were performed. Quality control and sterility testing were performed to check the quality of medium. Cary-Blair transport media was checked for its viability for one week by inoculating control strains and was found successful to store *Salmonella* or *Shigella* species up to one week at 2–8˚C. Reference strains of *Salmonella* species (ATCC 13076) and *Shigella* species (ATCC 12022) were used as a quality control for culture and susceptibility testing throughout the study. All reference strains were obtained from Dessie Health Research Regional laboratory.

## Statistical analysis

The data generated were entered in to Microsoft-Excel spreadsheet 2010 (Microsoft Cop., USA) and imported to be analyzed by Statistical Package for Social Sciences (SPSS) version 22.0. (IBM, USA). Descriptive statistics was used the compute percentages. Binary logistic regression was used to show the association of each variable with the dependent variable. Moreover, a multivariate analysis was computed to identify factors that independently influence the occurrence of the dependent variable. P-value <0.05 with 95% confidence interval was considered to declare statistical significance.

## Ethical considerations

Ethical clearance was obtained from Research and Ethics Review Committee of the Department of Medical Laboratory Sciences of Addis Ababa University. Moreover, prior to commencing the study, a written informed consent was obtained from each participant. Study participants aged less than 18 years were asked an assent and written consent was taken from their parents or guardians. Confidentiality and any special data security requirements were maintained and assured. Results of the laboratory examinations were informed to physicians and the participants got their results and treatment duly as required.

## Results

### Socio-demographic characteristics

In this study, a total 354 study participants (112 HIV infected and 242 non-infected diarrheic patients) were investigated during the study period. The study participants were enrolled from

three health facilities; of them 158 (52.2%) were from Dessie Referral Hospital, 48 (13.6%) from Buanbua Wuha Health Center, and 121 (34.2%) from Dessie Health Center with 100% response rate. Their age ranged from 15 to 86years, with mean age of 35.33±13.11 years. Distribution by sex revealed predominance of female cases in HIV infected 76(67%) and males 127 (52.5%) in HIV non infected cases. Majority of the HIV infected (73%) and non-infected (65.7%) were urban dwellers (Table 1).

### Etiologic agents of diarrhea

The overall prevalence of enteric pathogens in stool samples of diarrheal patients was 100 (28.2%). Among these 76 (21.5%) were intestinal parasites whereas 24 (6.8%) were bacterial infections. The prevalence of intestinal parasite and bacterial infection among HIV infected was 26 (23.2%) and 8 (7.1%), respectively. Of the bacterial isolates, the predominately isolated bacteria were *Salmonella* 6 (5.4%) (Table 2). While the most prevalent parasite was *C. parvum* 9 (8.0%), followed by *E. histolytica/dispar* 8 (7.1%), *G. lamblia* 4 (3.6%) and 1 (0.9%) co-infection of *C. parvum* and *Cyclospora cayetanensis*.

The prevalence of intestinal parasites and bacterial infection among HIV non-infected diarrheal patients was 50 (20.7%) and 16 (6.6%), respectively. The most prevalent intestinal

**Table 1. Socio-demographic characteristics of HIV infected and non-infected diarrheal patients (n = 354) from selected health facilities of Dessie Town, Northeast Ethiopia, from January to March 2018.**

| | | HIV status | | Total n (%) |
|---|---|---|---|---|
| | | HIV positive n (%) | HIV negative n (%) | |
| **Health facilities** | Dessie R Hospital | 67 (59.8) | 118 (77.7) | 185 (52.2) |
| | Buanbua Wuha HC | 10 (8.9) | 38 (15.7) | 48 (13.6) |
| | Dessie HC | 35 (31.3) | 86 (35.5) | 121 (34.2) |
| **Age (years)** | 15–24 | 19 (16.9) | 66 (27.3) | 85 (24) |
| | 25–34 | 33 (29.5) | 62 (25.6) | 95 (26.8) |
| | 35–44 | 38 (33.9) | 55 (22.7) | 93 (26.2) |
| | >44 | 22 (19.6%) | 59 (24.4) | 81 (22.8) |
| **Sex** | Male | 36 (32.1) | 127 (52.5) | 163 (46) |
| | Female | 76 (67.8) | 115 (47.5) | 191 (54) |
| **Residence** | Urban | 82 (73.2) | 159 (65.7) | 241 (68) |
| | Rural | 30 (26.8) | 83 (34.3) | 113 (31.9) |
| **Education status** | Illiterate | 53 (47.3) | 61 (25.2) | 114 (32.2) |
| | Primary | 35 (31.3) | 54 (22.3) | 89 (25.1) |
| | Secondary | 17 (15.2) | 65 (26.9) | 82 (23.2) |
| | College/ University | 7 (6.3) | 62 (25.6) | 69 (19.4) |
| **Occupation** | Civil servant | 20 (17.9) | 44 (18.2) | 64 (18) |
| | Private | 32 (28.6) | 64 (25.4) | 96 (27) |
| | Unemployed | 49 (43.6) | 84 (34.7) | 133 (37.5) |
| | Farmer | 11 (9.8) | 50 (20.7) | 61 (17.2) |
| **Monthly family income (Eth birr)** | < 500 | 37 (33) | 72 (29.6) | 109 (30.8) |
| | 501–1000 | 28 (25) | 35 (14.5) | 63 (17.8) |
| | 1001–1500 | 9 (8.0) | 44 (18.2) | 53 (15.0) |
| | 1501–2000 | 17 (15.2) | 32 (13.2) | 49 (13.8) |
| | >2000 | 21 (18.6) | 59 (24.4) | 80 (22.6) |

R = Referral; HC = Health center; Eth birr = Ethiopian Birr (equivalent to 1USD = 31.66 Eth birr at the time of the study)

**Table 2. Prevalence of bacterial pathogens and intestinal parasites among HIV infected and non-infected diarrheal patients (n = 354) in selected health facilities in Dessie Town, Northeast Ethiopia, from January to March 2018.**

| Enteric Pathogen | HIV positive (n = 112) No. (%) | HIV negative (n = 242) No. (%) | Total (n = 354) No. (%) |
|---|---|---|---|
| **Bacteria** | **8 (7.1)** | **16 (6.6)** | **24 (6.8)** |
| Salmonella species | 6 (5.4) | 11 (4.5) | 17 (4.8) |
| Shigella species | 2 (1.8) | 5 (2.1) | 7 (2.0) |
| **Intestinal parasites** | **26 (23.2)** | **50 (20.7)** | **76 (21.5)** |
| E. histolytica/dispar | 8 (7.1) | 39 (16.1) | 47 (13.3) |
| G. lamblia | 4 (3.6) | 8 (3.3) | 12 (3.4) |
| Taenia species | 2 (1.8) | 0 (0) | 2 (0.56) |
| C. parvum | 9 (8.0) | 0 (0) | 9 (2.5) |
| C. parvum & C. cayetanensis | 1 (0.9) | 0 (0) | 1 (0.3) |
| A. lumbricoides | 0 (0) | 2 (0.8) | 2 (0.56) |
| E. vermicularis | 2 (1.8) | 1 (0.4) | 3 (0.8) |
| **Total** | **34 (30.4)** | **66 (27.3)** | **100 (28.2)** |
| **Co-infections** | **3 (2.8)** | **1 (0.4)** | **4 (1.1)** |
| Shigella & C. parvum | 2 (0.6) | 0 (0) | 2 (0.6) |
| Salmonella & A. lumbricoides | 0 (0) | 1 (0.3) | 1 (0.3) |
| Salmonella & Taenia species | 1 (0.3) | 0 (0) | 1 (0.3) |

parasites in this group of patients were *E. histolytica/dispar* 39 (16.1%) and *G. lamblia* 8 (3.3%) (Table 2).

## Associated risk factors of *Salmonella* and *Shigella* infection

In this study, 14 independent variables were considered during the bivariate analysis of risk factors for the two enteric pathogens in diarrheal patients. Relatively higher prevalence of bacteria was found among the age group of 15–24 years, educational status of illiterates, in farmers, and among participants who do not utilize latrine, who drink river or spring water and those who had mucoid plus bloody diarrhea. In multivariate analysis, patients who had bloody plus mucoid diarrhea [AOR = 5.982, 95%CI: (1.41, 68.833), P = 0.026], who did not utilize latrine [AOR = 6.407, 95%CI: (1.139, 36.024), P = 0.035] and those who drink river or spring water sources [AOR = 8.641, 95%CI: (1.983, 37.656), P = 0.004] had a statistically significant independent association with intestinal *Salmonellosis* and *Shigellosis* regardless of HIV status (Table 3).

## Associated risk factors of parasitic infection

In this study, 12 independent variables were considered during the bivariate analysis of risk factors for intestinal parasites among diarrheal patients. Relatively higher prevalence of intestinal parasites was found among study participants who were in the age group of 15–24 years, rural residents, illiterate, unemployed, and who did not utilize latrine, who drink river or spring water, had monthly income below 500 birr and duration of diarrhea lasting 6–10 days. In multivariate analysis, participants who did not have formal education (were illiterate) [AOR = 5.8, 95%CI: (1.62, 21.04), P = 0.007], had primary level education [AOR = 3.7, 95%CI: (1.00, 13.744), P = 0.05], had 6–10 days of diarrhea [AOR = 2.039, 95%CI: (1.094, 3.80), P = 0.025], CD4 level between 200–500 cells/mm$^3$ [AOR = 6.48, 95%CI: (2.144, 19.592), P = 0.001] and those who did not wash their hand properly with soap [AOR = 3.02,

**Table 3. Associations of risk factors with bacterial agents among study participants (n = 354) attending selected health facilities in Dessie Town, Northeast Ethiopia, January to March 2018.**

| Variable | Bacterial infection | | COR (95% C.I) | P value | AOR (95%CI) | P Value |
|---|---|---|---|---|---|---|
| | Positive | Negative | | | | |
| **Age (years)** | | | | | | |
| 15–24 | 9 (10.6) | 76 (89.4) | 1 | | | |
| 25–34 | 6 (6.3) | 89 (93.7) | 0.569 (0.56–1.67) | 0.31 | | |
| 35–44 | 3 (3.2) | 90 (97.8) | 0.281 (0.28-.074) | 0.06 | | |
| >44 | 6 (7.4) | 75 (92.6) | 0.676 (0.23–1.99) | 0.48 | | |
| **Sex** | | | | | | |
| Male | 10 (6.0) | 153 (94.) | 1 | | | |
| Female | 14 (7.3) | 177 (92.7) | 1.2 (0.523–2.802) | 0.656 | | |
| **Residence** | | | | | | |
| Urban | 11 (4.6) | 230 (95.4) | 1 | | 1 | |
| Rural | 13 (11.5) | 100 (88.5) | 2.718 (1.178–6.275) | 0.019 | 0.271 (0.58–1.257) | 0.95 |
| **Occupation** | | | | | | |
| Civil servant | 1 (1.6) | 63 (98.4) | 1 | | 1 | |
| Privet employee | 4 (4.2) | 92 (95.8) | 2.74 (0.29–25.85) | 0.37 | 0.795 (0.057–11.17) | 0.865 |
| Unemployed | 12 (9.0) | 121 (91.0) | 6.25 (0.79–49.15) | 0.08 | 0.682 (0.038–12.15) | 0.794 |
| Farmer | 7 (11.5) | 54 (88.5) | 8.20 (0.97–68.49) | 0.05 | 0.095 (0.004–2.446) | 0.156 |
| **Education status** | | | | | | |
| Illiterate | 13 (11.4) | 101 (88.6) | 8.8 (1.12–68.47) | 0.04 | 2.62 (0.127–54.06) | 0.534 |
| Primary | 5 (5.6) | 84 (94.4) | 4.05 (0.46–35.48) | 0.20 | 1.13 (0.058–22.16) | 0.935 |
| Secondary | 5 (6.1) | 77 (93.9) | 4.42 (.50–38.74) | 0.18 | 3.16 (0.185–54.12) | 0.427 |
| Higher education | 1 (1.4) | 68 (98.6) | 1 | | 1 | |
| **Monthly Family Income (ETB)** | | | | | | |
| <500 | 14 (12.8) | 95 (87.2) | 3.78 (1.05–13.6) | 0.04 | 1.974 (0.24–18.224) | 0.549 |
| 501–1000 | 4 (6.3) | 59 (93.7) | 1.74 (0.375–8.07) | 0.47 | 0.765 (0.082–7.152) | 0.814 |
| 1001–1500 | 1 (1.9) | 52 (98.1) | 0.49 (0.05–4.876) | 0.55 | 0.480 (0.032–7.261) | 0.597 |
| 1501–2000 | 2 (4.0) | 47 (96.0) | 1.09 (0.176–6.78) | 0.92 | 0.611 (0.056–6.644) | 0.686 |
| >2000 | 3 (3.8) | 77 (96.2) | 1 | | 1 | |
| **Diarrhea duration (days)** | | | | | | |
| 1–5 | 15 (5.8) | 245 (94.2) | 1 | | | |
| 6–10 | 8 (9.6) | 75 (90.4) | 1.74 (0.71–4.27) | 0.22 | | |
| >10 | 1 (9.1) | 10 (90.9) | 1.6 (0.196–13.62) | 0.65 | | |
| **Diarrhea consistency** | | | | | | |
| Watery | 11 (4.4) | 237 (95.6) | 1 | | 1 | |
| Mucoid | 7 (13.2) | 46 (86.8) | 3.239 (1.208–8.902) | 0.02 | 1.983 (0.636–6.182) | 0.238 |
| Bloody | 2 (6.3) | 30 (93.70) | 1.436 (0.304–6.793) | 0.65 | 0.532 (0.094–3.017) | 0.476 |
| Blood + mucoid | 4 (19.0) | 17 (81.0) | **5.07 (1.46–17.6)** | 0.11 | **5.982 (1.41–68.833)** | **0.026***|
| **HIV status** | | | | | | |
| Yes | 8 (7.1) | 104 (92.9) | 1.087 (0.451–2.619) | 0.853 | | |
| No | 16 (6.6) | 226 (93.4) | 1 | | | |
| **CD4 Level** | | | | | | |
| <200cells/mm$^3$ | 1 (20) | 4 (80) | 3.778 (0.339–42.154) | 0.280 | 1.974 (0.24–12.254) | 0.749 |
| 200-500cells/mm$^3$ | 4 (11.8) | 30 (88.2) | 3.022 (0.637–14.344) | 0.164 | 0.765 (0.082–7.142) | 0.814 |
| >500cells/mm$^3$ | 8 (7.1) | 104 (92.9) | 1 | | **1** | |
| **Hand wash before meal** | | | | | | |
| Hand wash without soap | 18 (13.8) | 112 (86.2) | 5.839 (1.459–17.618) | 0.00 | 1.624 (0.345–7.638) | 0.539 |

*(Continued)*

**Table 3.** (Continued)

| Variable | Bacterial infection | | COR (95% C.I) | P value | AOR (95%CI) | P Value |
|---|---|---|---|---|---|---|
| | Positive | Negative | | | | |
| Hand wash with soap | 6 (2.7) | 218 (97.3) | 1 | | 1 | |
| Latrine utilization | | | | | | |
| Utilized | 8 (2.8) | 280 (97.2) | 1 | | 1 | |
| Not utilized | 16 (24.2) | 50 (75.8) | **11.2 (4.552–27.56)** | **0.00** | **6.407 (1.139–36.024)** | **0.035**[*] |
| Drinking water source | | | | | | |
| Pipe | 7 (2.4) | 279 (97.6) | 1 | | 1 | |
| Not pipe (river) | 17 (25) | 51 (75.0) | **13.286 (5.245–33.654)** | **0.00** | **8.641 (1.983–37.656)** | **0.004**[*] |

Note:

[*]Statistically significant at P<0.05.

AOR = adjusted odds ratio, COR = crude odds ratio, 1 = reference group, 95% CI = 95% confidence interval.

95%CI: (1.5, 6.23), P = 0.003] had statistically significant association with intestinal parasites (Table 4).

### Antimicrobial susceptibility pattern of bacterial isolates

All bacterial isolates from diarrheal patients (Table 5) were 100% susceptible to ceftriaxone, 95.4% to ciprofloxacin while 100% resistant to ampicillin and amoxicillin. Moreover, tetracycline, trimethoprim sulfamethoxazole/cotrimoxazole, chloramphenicol and nalidixic acid showed variable degrees of antimicrobial resistance as shown in Table 5. When data is disaggregated by species type, *Salmonella* isolates were 100% resistant to Ampicillin and amoxicillin while being sensitive to ciprofloxacin and ceftriaxone. On the other hand, *Shigella* species isolates were 100% resistant to ampicillin but susceptible to ciprofloxacin (95.6%) and ceftriaxone (100%) (Table 5).

### Multiple drug resistance patterns of the isolates

Overall, 24 (100%) bacterial isolates were resistant to at least one antimicrobial agent (Table 6). Multidrug resistance (MDR = Non-susceptible to ≥1 agent in ≥3 antimicrobial categories) [26] was seen in 19 (79.2%) of the isolates. About 76.5% of *Salmonella* isolates and 85.7% of *Shigella* isolates showed multidrug resistance for the tested antimicrobial drugs (Table 6).

## Discussion

In this study the magnitude as well as predicting factors of *Salmonella*, *Shigella* and intestinal parasitic infections were determined among HIV infected and non-infected diarrheal patients at selected health facilities in Dessie Town, Northeast Ethiopia. Accordingly, intestinal parasites were detected in 23.2% of HIV infected and 21.7% HIV non-infected participants while bacterial infections were seen in 7.1% and 6.6% of HIV infected and non-infected patients, respectively.

The finding of 1.8% *Shigella* prevalence in HIV infected patients of the present study was relatively close to findings of 1.1% from Jimma, southwest Ethiopia [14]. However, our finding was lower compared to the 3.7% from Pune India [8] and 3.5% reported from studies done in Gondar, northwest Ethiopia [27] and 5.3% in Nigeria [28]. A study from Kampala, Uganda reported a prevalence rate of 11.1% in HIV infected children with acute diarrhea [29]. The variations noted might be due to age group, socioeconomic factor, the nature of the public water

**Table 4. Bivariate and multivariate logistic regression analyses of factors associated with intestinal parasites among diarrheal patients (n = 354) attending selected health facilities in Dessie Town, Northeast Ethiopia, from Januarys to March 2018.**

| Variable | Intestinal parasite | | COR(95% C.I) | P Value | AOR(95% C.I) | P value |
|---|---|---|---|---|---|---|
| | Positive | Negative | | | | |
| **Age Group** | | | | | | |
| 15–24 | 23 (27.1) | 62 (72.9) | 1 | | | |
| 25–34 | 15 (15.8) | 80 (84.2) | 0.51 (0.24–1.05) | 0.067 | | |
| 35–44 | 17 (18.3) | 76 (81.7) | 0.60 (0.29–1.23) | 0.163 | | |
| >44 | 21 (25.9) | 60 (74.1) | 0.94 (0.47–1.88) | 0.869 | | |
| **Residence** | | | | | | |
| Urban | 42 (17.4) | 199 (82.6) | 1 | | 1 | |
| Rural | 34 (30.0) | 79 (70.0) | 2.04 (1.21–3.44) | 0.007 | 0.947 (0.48–1.866) | 0.874 |
| **Occupation** | | | | | | |
| Civil servant | 5 (7.8) | 59 (92.2) | 1 | | 1 | |
| Private | 15 (15.6) | 81 (84.4) | 2.2 (0.75–6.35) | 0.151 | 1.078 (0.332–3.495) | 0.901 |
| Unemployed | 32 (24.0) | 101 (76.0) | 3.7 (1.4–10.12) | 0.009 | 1.093 (0.335–3.563) | 0.882 |
| Farmer | 24 (24.0) | 76 (76.0) | 7.6 (2.68–21.8) | 0.000 | 2.421 (0.553–10.59) | 0.240 |
| **Educational status** | | | | | | |
| Higher | 3 (4.3) | 66 (95.7) | 1 | | 1 | |
| Illiterate | 37 (32.5) | 77 (67.5) | 10.6 (3.2–35.9) | 0.000 | 5.8 (1.62–21.04) | 0.007* |
| Primary | 21 (23.6) | 68 (76.4) | 6.79 (1.9–23.86) | 0.003 | 3.7 (1.00–13.744) | 0.050* |
| Secondary | 15 (18.3) | 67 (81.7) | 4.93 (1.36–17.8) | 0.015 | 3.5 (0.95–13.744) | 0.060 |
| **Monthly Income (ETB)** | | | | | | |
| <500 | 36 (36.3) | 73 (73.7) | 5.14 (2.15–12.3) | 0.000 | 1.738 (0.60–5.031) | 0.308 |
| 501–1000 | 14 (22.2) | 49 (77.8) | 2.98 (1.12–7.91) | 0.028 | 1.276 (0.418–3.897) | 0.669 |
| 1001–1500 | 9 (17.0) | 44 (81.0) | 2.13 (.74–6.13) | 0.160 | 1.072 (0.337–3.411) | 0.907 |
| 1501–2000 | 10 (20.4) | 39 (79.6) | 2.67 (0.94–0.57) | 0.064 | 1.667 (0.543–5.118) | 0.372 |
| >2000 | 7 (8.8) | 73 (91.2) | 1 | | 1 | |
| **Diarrhea duration (days)** | | | | | | |
| 1–5 | 45 (17.3) | 215 (82.7) | 1 | | 1 | |
| 6–10 | 29 (34.9) | 54 (65.1) | 2.56 (1.48–4.47) | 0.001 | 2.039 (1.094–3.80) | 0.025* |
| >10 | 2 (18.2) | 9 (81.8) | 1.06 (0.222–5.08) | 0.940 | 1.629 (0.312–8.50) | 0.563 |
| **Diarrhea consistency** | | | | | | |
| Watery | 45 (18.1) | 203 (81.9) | 1 | | 1 | |
| Mucoid | 17 (32.0) | 36 (68.0) | 2.13 (1.10.126) | 0.025 | 1.249 (0.581–2.686) | 0.568 |
| Bloody | 9 (28.1) | 23 (71.9) | 1.8 (0.765–4.07) | 0.183 | 1.235 (0.46–3.314) | 0.675 |
| Blood + mucoid | 5 (23.8) | 16 (76.2) | 1.4 (0.49–4.048) | 0.523 | 1.818 (0.576–5.735) | 0.308 |
| **HIV status** | | | | | | |
| Yes | 26 (23.2) | 86 (76.8) | 1.161 (0.678–1.98) | 0.587 | | |
| No | 50 (20.7) | 192 (79.3) | 1 | | | |
| **CD4 Level** | | | | | | |
| <200cells/mm³ | 3 (42.9) | 4 (57.1) | 4.575 (0.88–23.57) | 0.69 | 4.058 (0.701–23.499) | 0.118 |
| 200-500cells/mm³ | 13 (38.2) | 21 (61.8) | 3.776 (1.44–9.883) | 0.007 | 6.48 (2.144–19.592) | 0.001* |
| >500cells/mm³ | 10 (14.1) | 61 (85.9) | 1 | | 1 | |
| **Hand wash before meal** | | | | | | |
| Hand wash with soap | 29 (12.9) | 195 (87.1) | 1 | | 1 | |
| Hand wash without soap | 47 (36.2) | 83 (63.8) | 3.8 (2.24–6.47) | 0.001 | 3.02 (1.5–6.23) | 0.003* |
| **Latrine utilization** | | | | | | |
| Utilize | 53 (18.4) | 235 (81.6) | 1 | | 1 | |

*(Continued)*

**Table 4.** (Continued)

| Variable | Intestinal parasite | | COR(95% C.I) | P Value | AOR(95% C.I) | P value |
|---|---|---|---|---|---|---|
| | Positive | Negative | | | | |
| Not utilize | 23 (34.8) | 43 (65.2) | 2.37 (1.32–4.27) | 0.004 | 1.86 (0.501–3.836) | 0.529 |
| **Water source** | | | | | | |
| Pipe | 56 (19.6) | 230 (80.4) | 1 | | | |
| Not pipe (river) | 20 (29.4) | 48 (70.6) | 1.7 (0.94–3.11) | 0.078 | | |

Note:

*Statistically significant at *P*<0.05.

**AOR** = adjusted odds ratio, **COR** = crude odds ratio, **1** = reference group, **95% CI** = 95% confidence interval

supply and seasonal variations among the various studies. Nonetheless, the observed prevalence rates in the current and other studies are expected in diarrheic patients.

The current study detected *Salmonella* species in 6 (5.4%) HIV infected diarrhea patients. This finding was in line with a report from Kampalla Uganda (4.3%) [29] but higher than studies done in Nigeria (1.3%) [28] and Limma Peru (2%) [30]. However; our finding was lower than studies from Jimma southwest Ethiopia (10.8%) [14]. It is likely that differences in hygienic practices and access to clean water could explain the observed variations among the studies than differences in immune status. This is supported by the findings of no significant difference between HIV infected and non-infected participants of the current study.

The prevalence of bacterial infection among HIV non-infected patients was 16 (6.6%). Among these *Salmonella* accounts 11(4.5%); this finding was higher than studies done in Hawassa, southern part of Ethiopia (2.5%) [18] and Gaborone, Botswana (3.0%) [31]. Moreover, this finding was higher than two studies done in Gondar which reported 1.08% [27] and 1.04% [19] while lower than the prevalence rates from Harar 11.5% [20] and Butajira 10.5% [32]. The isolation rate of *Shigella* species among HIV negative study participants in our study 5 (2.1%) was relatively comparable to the studies conducted in Jimma 2.3% [33] and Kampala, Uganda 3.5% [29]. Whereas the finding was lower when compared to previous studies conducted in different parts of Ethiopia including North west Ethiopia 8.7% [27], Harar 6.7% [20], Gondar 16.9% [19]; and in Gaborone, Botswana 21% [31]. In general, reports on the

**Table 5. Antimicrobial susceptibility pattern of *Salmonella* and *Shigella* isolates among diarrheic patients attending selected health facilities of Dessie Town, Northeast Ethiopia from January to March 2018.**

| Antibiotics | Salmonella species N = 17 (N/%) | | | Shigella species N = 7 (N/%) | | | Total N = 24 (N/%) | | |
|---|---|---|---|---|---|---|---|---|---|
| | S | I | R | S | I | R | S | I | R |
| AMP(10 µg) | 0(0.0) | 0(0.0) | 17 (100) | 0(0.0) | 0(0.00) | 7(100) | 0(0.0) | 0(0.0) | 24(100) |
| AMX (20 µg) | 0(0.0) | 0(0.0) | 17(100) | 0(0.0) | 0(0.0) | 7(100) | 0(0.0) | 0(0.0) | 24(100) |
| TTC (30 µg) | 9(52.9) | 6(35.4) | 2(11.8) | 4(57.1) | 1(14.3) | 2(28.6) | 13(54.2) | 7(29.1) | 4(16.7) |
| CAF (30 µg) | 11(64.7) | 3(17.6) | 3(17.6) | 5(71.4) | 1(14.3) | 1(14.3) | 16(66.7) | 4(16.7) | 4(16.7) |
| COT (25 µg) | 10(58.8) | 3(17.6) | 4(23.5) | 4(53.1) | 0(0) | 3(46.9) | 14(58.3) | 3(12.5) | 7(29.2) |
| NAL (30 µg) | 14(82.4) | 2(11.8) | 1(5.9) | 4(53.1) | 1(14.3) | 2(28.6) | 17(70.8) | 4(16.7) | 3(12.5) |
| CRX (30 µg) | 17(100) | 0(0.0) | 0(0.0) | 7(100) | 0(0.0) | 0(0.0) | 24(100%) | 0(0.0) | 0(0.0) |
| CIP (5 µg) | 17(100) | 0(0.0) | 0(0.0) | 6(85.7) | 0(0.0) | 1(14.3) | 23(95.8) | 0(0.0) | 1(4.2) |

S = Susceptibility, I = Intermediate, R = Resistant, AMP = Ampicillin, AMX = Amoxicillin, TTC = Tetracycline, CAF = Chloramphenicol, COT = Cotrimoxazole, NAL = Nalidixic acid, CRX = Ceftriaxone, CIP = Ciprofloxacin

**Table 6. Multi drug resistance pattern of *Salmonella* and *Shigella* isolates from diarrheal patients in selected health facilities of Dessie Town, Northeast Ethiopia, from January to March, 2018.**

| Bacterial isolates | Total | Antimicrobial resistance pattern | | | | | | | |
|---|---|---|---|---|---|---|---|---|---|
| | | Ro | R1 | R2 | R3 | R4 | R5 | R6 | MDR |
| *Salmonella* | 17(70.8) | 0(0.0) | 0(0.0) | 4(23.5) | 4(23.5) | 7(41.1) | 2(11.8) | 0(0.0) | 13(76.5)[a] |
| *Shigella* | 7(29.2) | 0(0.0) | 0(0.0) | 1(14.3) | 2(28.6) | 3(37.5) | 0(0.0) | 1(14.3) | 6(85.7)[b] |
| Total | 24(100) | 0(0.0) | 0(0.0) | 5(20.8) | 6(25) | 10(41.7) | 2(8.3) | 1(4.2) | 19(79.2)[c] |

Ro = No antibiotic resistance, R1 = Resistance to one, R2 = Resistance to two, R3 = Resistance to three, R4 = Resistance to four, R5 = Resistance to five, R = Resistance to six and more drugs, MDR = Multi-drug resistant,

[a] Percent is computed from total number of *Salmonella*,

[b] Percent is computed from total number of *Shigella* isolates,

[c] Percent is computed from total number of isolates based on which MDR definition is applied.

magnitude of these two enteric pathogens vary among studies. The discrepancy might be attributable to differences in awareness of the people about personal and environmental hygiene among different places; in the nature of the water supply system; access to health facilities and awareness regarding food borne infectious diseases.

On the other hand, intestinal parasites were detected in 23.2% of the HIV/AIDS patients and 20.6% of the HIV negative patients. Our finding was low compared to studies conducted from Addis Ababa (50.3%, 41.1%) [34], southwest Ethiopia (44.8%, 37.8%) [35], Bahir Dar (80.3%, 33.3%) [36], Hawassa (59.8%, 48.8%) [37], Cameroon (59.5%, 9.32%) [6] and Limma Peru (55%, 21%) [30] in Latin America, for HIV positive and HIV negative participants, respectively. The reports of Cameroon were low in HIV negatives and that of Lima Peru were in line with our finding. The low rate of isolation as observed in the present study might be due to the increasing awareness of the people about personal and environmental hygiene made by the health institutions and other partners.

In this study, the prevalence of cryptosporidium species was 9 (8.0%). This result was nearly similar with findings from selected ART centers of Adama, Afar and Dire-Dawa (8%) [38], southwestern Ethiopia (11%) [35], Cameroon (12.6%) [39] and Pune India (14.8%) in Asia continent [8]. This result was higher compared to reports from Dessie, Ethiopia (1.5%) [40]. However, our finding was lower compared to several studies conducted in Addis Ababa (25%) [34], Bahir Dar (43.6%) [36], Hawassa (20%) [37], in Ethiopia, Cameroon (19%) [6] and Lima Peru (20%) in Latin America [30]. This low prevalence in this study might be due to early starting of ART, geographic differences and time gap between studies; nowadays there is a better awareness of patients about ART treatment, intestinal parasite infection and their causes.

The current study tried to identify risk factors associated with *Salmonella* and *Shigella* infections. Patients having bloody mucoid diarrhea with pus were about 6 times more likely [AOR = 5.982, 95%CI: (1.41, 68.833), P = 0.026] to harbor these enteric pathogens than those with watery diarrhea. This finding agrees with similar reports from Mekelle [41] and Harar [20]. This is possibly due to the ability of those bacteria to destruct tissue, invade and replicate in cells lining the colon and rectum. The bacteria serotype is also another cause of bloody plus mucoid diarrhea in stool samples.

Latrine usage and source of drinking water were identified as predisposing factors of bacterial infection as evidenced by multivariate analysis of this study. Patients who did not utilize latrine had been about 6 times more likely to have bacterial infections of *Salmonella* and *Shigella* [AOR = 6.407 95%CI: (1.139, 36.024), (P = 0.035)] than patients who did. This could be due to open space defecation (ineffective feces disposal) which attracts flies and other insects

that can transmit those diseases. This finding was supported by a study done in Mekelle which showed 7.5 times more risk among those who did not have toilet in their home [41].

Drinking water source was another factor that exposed patients to *Salmonella* and *Shigella* infections. Patients who drink river or spring water (untreated water) were about 9 times more likely to have *Salmonella* and *Shigella* infections [AOR = 8.641, 95%CI:(1.983, 37.656), (P = 0.004)] than those who drink treated pipe water. It is obvious that the bacteria mainly transmitted person-to-person by the feco-oral route, primarily by people with unhygienic hands and through drinking contaminated water. Unhygienic practices accounting for the higher likelihood of infection by these bacterial agents is also documented by the study in Mekelle city [41]. This is in line with a well-established fact that diarrhea is a global problem in areas with limited access to safe water and where poor hygiene and sanitation are commonly practiced [2]. In this study, illiterate participants were 5.8 times more likely and those at primary education were 3 times more likely to have intestinal parasites than those who attained higher education. People with no formal education and those at primary level could have limited awareness about the transmission route of intestinal parasites.

On the other hand, patients who had diarrhea elapsing for 6–10 days had more than twofold significant association with intestinal parasite positivity [AOR = 2.039, 95%CI:(1.094, 3.80), (P = 0.025)]. Moreover, as can be expected, patients who did not wash hands properly with soap before meal had 3 times higher risk of having intestinal parasitic infection [AOR = 3.02, 95%CI:(1.5, 6.23) (P = 0.003)] than patients who wash their hands properly with soap.

Finally, drug susceptibility pattern analysis in the current study revealed resistance of all bacterial strains to at least two drugs. Of note, ampicillin, amoxicillin, cotrimoxazole and tetracycline were not effective drugs. The main reason could be the frequent use of these antibiotics in the country. Among the seventeen *Salmonella* species isolates, the overall rate of resistance was high for Amoxicillin and Ampicillin (100%), Tetracycline (47%), Cotrimoxazol (41%), Chloramphenicol (35.2%) and Nalidixic acid (17.6%) whereas 17 *Salmonella* species were 100% sensitive to Ceftriaxone and Ciprofloxacin. This finding was comparable with the results of Gondar [27] which demonstrated resistance of Ampicillin (75%), Amoxicillin (100%), Cotrimoxazole (50%), Nalidixic acid (25.0%). Such high resistance rates were reported from other parts of Ethiopia like Jimma [14] (100%) Ampicillin, (47%) Tetracycline, (26%) Nalidixic acid, and Harar [20] (both Ampicillin and Amoxicillin were 100% resistant). In contrast to our findings, the study from Sudan showed 100% sensitivity of their isolates to chloramphenicol and tetracycline, while 64% to ampicillin [42], and all isolates of *Salmonella* showed susceptibility to all antibiotics like Ampicillin, Tetracycline, Cotriamoxole and Chloramphenicol as reported in Gondar [19]. The time lapse between the two Ethiopian studies could partly explain the difference in the sensitivity pattern.

In this study, *Shigella* species showed the highest resistance to Ampicillin 7 (100%), Cotrimoxazole 3 (46%) and Amoxicillin 7 (100%). This finding has to be interpreted with care, since the numbers of isolates are small. Nonetheless, similar findings have been reported by studies done at Harar [20], Gondar [27], and Mekelle [41] which demonstrated resistance rates between 88% to 100% to both Ampicillin and Amoxicillin. This could be due to the over use of these drugs for many years. Therefore, according to this study, Ampicillin, Tetracycline, Amoxicillin and SXT are no longer effective for the treatment of *Shigellosis* and *Salmonellosis* in the study area. On the other hand, lower resistance (higher rate of sensitivity) rates were observed against Ceftriaxone and Ciprofloxacin which is an indicative of possible use of these drugs as an empiric therapy particularly in the study area. The possible justification for such low level resistance might be attributable to infrequent prescription of these drugs. Both the Ethiopian [23] and WHO recommendations [21, 22] placed Ciprofloxacin as a first-line treatment option in the management of *Shigellosis*. Our study provides evidence reassuring that the

guideline would work well in our setting. Hence, Ciprofloxacin should be continued as a first line; and Ceftriaxone could be considered as alternative options in the treatment of *Salmonella* and *Shigella* infections.

## Conclusion and recommendation

The prevalence of intestinal parasite and bacterial infection is still high in HIV infected and non-infected patients, particularly *E.histolytica/dispar*, Cryptosporidium species and salmonella. Unhygienic practices, unclean water source, and having bloody and mucoid diarrhea are factors independently associated with enteric bacterial infections. On the other hand, educational status, CD4 level and hand washing habit with soap before meal are factors associated with intestinal parasite prevalence. Ampicillin, amoxicillin and trimethoprim sulfamethazine are no longer effective for the treatment of *Salmonella* and *Shigella* species. *Salmonella* and *Shigella* isolates in this study are highly sensitive to Ciprofloxacin and ceftriaxone and hence could be recommended in areas where there is no culture and antimicrobial sensitivity testing facility. Of the total isolates, 76.5% and 85.7% of them were MDRs *Salmonella* and *Shigella* species, respectively. Therefore, as much as possible, early identification of diarrheal etiologic agents and antimicrobial susceptibility testing should be practiced to select the suitable antimicrobial agent and prevent the emergence as well as spread of multi drug resistant bacterial strains. Moreover, attention needs to be given to clean environment, improvement of latrine utilization, availability of safe water sources as well as health education on the transmission of enteric pathogens and IPs. In addition, emerging modified techniques in ART clinic to improve diagnosis of patients with diarrhea should be practiced.

## Supporting information

**S1 Questionnaire. Survey of bacterial and parasitic profile of HIV infected and non-infected diarrheal patients in health facilities of Dessie, Northeast Ethiopia.** (DOCX)

## Acknowledgments

The authors would like to acknowledge Dessie Regional Health Research Laboratory for providing laboratory space and facilities to conduct the experiments. Health facilities in Dessie town and all study participants are gratefully acknowledged for their kind cooperation.

## Author Contributions

**Conceptualization:** Assefa Belay, Berhanu Seyoum, Aster Tsegaye.

**Data curation:** Assefa Belay, Mekuanent Alemu.

**Formal analysis:** Assefa Belay, Melaku Ashagrie, Mekuanent Alemu, Aster Tsegaye.

**Investigation:** Assefa Belay, Melaku Ashagrie, Mekuanent Alemu.

**Methodology:** Assefa Belay, Mekuanent Alemu, Aster Tsegaye.

**Project administration:** Assefa Belay, Berhanu Seyoum, Mekuanent Alemu, Aster Tsegaye.

**Resources:** Assefa Belay, Melaku Ashagrie, Mekuanent Alemu.

**Software:** Assefa Belay, Aster Tsegaye.

**Supervision:** Melaku Ashagrie, Berhanu Seyoum, Aster Tsegaye.

**Validation:** Assefa Belay, Melaku Ashagrie, Berhanu Seyoum, Mekuanent Alemu, Aster Tsegaye.

**Visualization:** Assefa Belay, Melaku Ashagrie, Aster Tsegaye.

**Writing – original draft:** Assefa Belay, Melaku Ashagrie, Berhanu Seyoum, Mekuanent Alemu, Aster Tsegaye.

**Writing – review & editing:** Assefa Belay, Melaku Ashagrie, Berhanu Seyoum, Mekuanent Alemu, Aster Tsegaye.

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
