## [Decision Letter · Decision Letter 0]

25 Sep 2020

PONE-D-20-27382

Prevalence of EntericPathogens,Intestinal Parasites and High Resistance rate of Bacterial Isolates to Commonly Prescribed Antibiotics in HIVInfected and Non-Infected Diarrheic Patients from selected Health Facilities in Dessie town,Northeast Ethiopia

PLOS ONE

Dear Dr. Tsegaye,

Thank you for submitting your manuscript to PLOS ONE. After careful consideration, we feel that it has merit but does not fully meet PLOS ONE’s publication criteria as it currently stands. Therefore, we invite you to submit a revised version of the manuscript that addresses the points raised during the review process.

A number of aspects of manuscript need improvement particularly outlining the objectives of statistical analysis and discussing the results rather than just comparing data to that reported by others.

We look forward to receiving your revised manuscript.

Kind regards,

Iddya Karunasagar

Academic Editor

PLOS ONE

Journal Requirements:

3.Thank you for stating the following in the Funding Section of your manuscript:

[No external funds were obtained; only institutional support from Dessie Regional Health Research

Laboratory.]

 [The author(s) received no specific funding for this work.]

5.We note that you have indicated that data from this study are available upon request. PLOS only allows data to be available upon request if there are legal or ethical restrictions on sharing data publicly. For information on unacceptable data access restrictions, please see http://journals.plos.org/plosone/s/data-availability#loc-unacceptable-data-access-restrictions.

6.Please upload a copy of Supporting Information S1 Questionnaire. which you refer to in your text on page 29.

7.We noticed you have some minor occurrence of overlapping text with the following previous publication(s), which needs to be addressed:

Belete MA. Bacterial Profile and ESBL Screening of Urinary Tract Infection Among Asymptomatic and Symptomatic Pregnant Women Attending Antenatal Care of Northeastern Ethiopia Region. Infect Drug Resist. 2020;13:2579-2592

https://doi.org/10.2147/IDR.S258379

In your revision ensure you cite all your sources (including your own works), and quote or rephrase any duplicated text outside the methods section. Further consideration is dependent on these concerns being addressed."

Additional Editor Comments (if provided):

The reviewer has pointed out number of aspects of the manuscript that need improvement. Please address all comments point by point.

Reviewers' comments:

Reviewer's Responses to Questions

**Comments to the Author**

1. Is the manuscript technically sound, and do the data support the conclusions?

Reviewer #1: Yes

2. Has the statistical analysis been performed appropriately and rigorously? 

Reviewer #1: Yes

3. Have the authors made all data underlying the findings in their manuscript fully available?

Reviewer #1: No

4. Is the manuscript presented in an intelligible fashion and written in standard English?

Reviewer #1: Yes

5. Review Comments to the Author

Reviewer #1: This is a well written manuscript that provides some useful data about pathogen prevalences and antimicrobial resistance in one area of Ethiopia. The methods seem good and well reported (the random selection of facilities is particularly commendable, as is the detailed description lab methods employed). Congratulations to the authors! I do have a few comments:

Major comments:

1) The authors need to discuss the recommended care of salmonella and shigella (both Ethiopian guidelines and WHO guidelines). For example, current WHO Shigella recommendation are to treat with Cipro. Your study suggest that guideline would work well in this region, which is reassuring. However, you then suggest Cipro as an “alternative” treatment, when in fact perhaps it should be the first line (as per guidelines)? I would suggest including the current treatment recommendation in the introduction and discussing how your results should inform these recommendation in the discussion.

2) I’m not sure that your conclusion that “the prevalence of these pathogens are high” (paraphrased) is valid. You only recruited children with diarrhea, so you were always going to see a high prevalence of enterics pathogens, and your discuss of other studies suggest that your prevalence is the same or lower than other studies have observed.

3) The discussion focus heavily on comparing your results with other work. While this is somewhat necessary for a discussion, many of the paragraphs do not appear to have an overarching point to the comparison. The discussion would be greatly improved by the authors make it clear to the reader what they should take away from each comparison. For example, in the second paragraph you compare prevalences of Shigella to other studies, what is your conclusion from this comparison?

4) In the statistical methods it would be better to identify which methods are answering which questions, i.e. you used descriptive methods to estimate the prevalences, and then model to associate exposures with the presence of pathogens. This would make it crystal clear to the reader what you are going to do.

5) You do not mention that you’re going to do a corelates of pathogen infection analysis in the introduction, you only mention looking at the prevalence. I was surprised by your correlates of infection analysis when I reached it in the results. Making this a clear goal of the paper in the introduction would be helpful.

Minor comments

1) There are some minor grammar errors, although the paper is generally well written:

Background, first paragraph, third line: “safe water is limited”

Background, fourth paragraph, seventh line: “5 times more bacteraemia”

Methods, first para, second line, “and two randomly”

Conclusion, fourth line, “On the other hand”

2) In the methods the sentence beginning “ A total of 354 HIV infected…” should be moved to the results.

3) In the antimicrobial susceptibility testing paragraph in the methods, you define the abbreviation CLSI two times.

6. PLOS authors have the option to publish the peer review history of their article (what does this mean?). If published, this will include your full peer review and any attached files.

Reviewer #1: No

---

## [Author Response · Author response to Decision Letter 0]

2 Nov 2020

Manuscript ID number:

PONE-D-20-27382

Title of paper:

Prevalence of Enteric Pathogens, Intestinal Parasites and High Resistance rate of Bacterial Isolates to Commonly Prescribed Antibiotics in HIV Infected and Non-Infected Diarrheic Patients from selected Health Facilities in Dessie town, Northeast Ethiopia

General

We thank all the reviewers for critically reviewing our manuscript which helps us to give better clarity to the paper and make it scientifically strong. We thank the PLOS ONE academic editors for their valuable comments and for giving us an opportunity to revise the manuscript. 

All the questions raised by the Reviewers and Editor have been addressed; and the manuscript is modified accordingly. Moreover, the requested Editorial corrections are addressed in both the revised manuscript and the response letter. Changes are shown with track changes in the file labeled ‘Revised Manuscript with Track Changes’. The point by point response is given below.

Editor Comments:

Author’s Response: 

 Thanks for the significant editorial comment. The revised manuscript is now updated and meets PLOS ONE's style requirements, including those for file naming.

Author’s Response: 

 Thank you for the comment. Copy of the questionnaire in both the original language and English is attached as Supporting Information.

3. We note that you have provided funding information that is not currently declared in your Funding Statement. However, funding information should not appear in the Acknowledgments section or other areas of your manuscript. We will only publish funding information present in the Funding Statement section of the online submission form. Please remove any funding-related text from the manuscript and let us know how you would like to update your Funding Statement. Currently, your Funding Statement reads as follows: [The author(s) received no specific funding for this work.]

Author’s Response: 

 Thank you and corrections are made as per the comment; the Funding Statement is amended and written as: “The authors received no specific funding for this work.” 

Moreover, the amended statement is included within the cover letter.

4. Please ensure that you have an ORCID iD and that it is validated in Editorial Manager.

Author’s Response: 

 Thank you for the valuable comment. Based up on the comment, pre-existing ORCID iD of the corresponding author is authenticated in the PLOS ONE Editorial Manager.

5. We note that you have indicated that data from this study are available upon request. PLOS only allows data to be available upon request if there are legal or ethical restrictions on sharing data publicly.

Author’s Response: 

 Thanks for the comment. Since the study included HIV positive patients, ethical restrictions do not allow us to share their data. However, the manuscript contains all the necessary information. 

6. Please upload a copy of Supporting Information S1 Questionnaire. which you refer to in your text on page 29.

Author’s Response: 

 Thanks, Supporting Information S1 Questionnaire is uploaded with the revised manuscript.

7. We noticed you have some minor occurrence of overlapping text with the following previous publication(s), which needs to be addressed. 

Author’s Response: 

 Thanks, comment well taken; we have tried to rephrase overlapping and duplicated texts as per the comments.

Reviewer #1 comments

1. The authors need to discuss the recommended care of salmonella and shigella (both Ethiopian guidelines and WHO guidelines). For example, current WHO Shigella recommendation are to treat with Cipro. Your study suggest that guideline would work well in this region, which is reassuring. However, you then suggest Cipro as an “alternative” treatment, when in fact perhaps it should be the first line (as per guidelines)? I would suggest including the current treatment recommendation in the introduction and discussing how your results should inform these recommendations in the discussion.

Author’s Response: 

 Thanks for the invaluable comment. We addressed this in both sections [Ref 21-23]

2. I’m not sure that your conclusion that “the prevalence of these pathogens are high” (paraphrased) is valid. You only recruited children with diarrhea, so you were always going to see a high prevalence of enterics pathogens, and your discuss of other studies suggest that your prevalence is the same or lower than other studies have observed.

Author’s Response: 

 Thank you for the substantial comment. We have revised as per the comment.

3. The discussion focuses heavily on comparing your results with other work. While this is somewhat necessary for a discussion, many of the paragraphs do not appear to have an overarching point to the comparison. The discussion would be greatly improved by the authors make it clear to the reader what they should take away from each comparison. For example, in the second paragraph you compare prevalences of Shigella to other studies, what is your conclusion from this comparison?

Author’s Response: 

 Comment well taken and after each comparison a concluding sentence is added, we thank you very much for this.

4. In the statistical methods it would be better to identify which methods are answering which questions, i.e. you used descriptive methods to estimate the prevalences, and then model to associate exposures with the presence of pathogens. This would make it crystal clear to the reader what you are going to do.

Author’s Response: 

 Thank you for the valuable comment. The statistical methods part is amended accordingly.

5. You do not mention that you’re going to do a corelates of pathogen infection analysis in the introduction, you only mention looking at the prevalence. I was surprised by your correlates of infection analysis when I reached it in the results. Making this a clear goal of the paper in the introduction would be helpful.

Author’s Response: 

 Thanks, and we have added a clear statement stating about the goal of the paper to do corelates of pathogen infection analysis and assess possible associated risk factors at the end of the introduction part of the revised manuscript. 

6. There are some minor grammar errors, although the paper is generally well written:

Background, first paragraph, third line: “safe water is limited”

Background, fourth paragraph, seventh line: “5 times more bacteraemia”

Methods, first para, second line, “and two randomly”

Conclusion, fourth line, “On the other hand”

Author’s Response: 

 Comments well taken and the manuscript corrected accordingly, thanks. 

7. In the methods the sentence beginning “ A total of 354 HIV infected…” should be moved to the results.

Author’s Response: 

 Thanks; we really appreciate the comment which helped us to notice a gap. Since the result section has the similar information, we rather felt the data collection section lacks about how the data was collected. Hence, we revised the sentence and included it under this section to make the reader aware from the beginning that data was collected from the study participants conveniently.

8. In the antimicrobial susceptibility testing paragraph in the methods, you define the abbreviation CLSI two times.

Author’s Response: 

 Comment addressed in the method section, thanks. 

Finally we thank you for your critical review of the paper.

Sincerely,

Aster Tsegaye, PhD

---

## [Decision Letter · Decision Letter 1]

23 Nov 2020

Prevalence of Enteric Pathogens, Intestinal Parasites and Resistance Profile of Bacterial Isolates among HIV Infected and Non-Infected Diarrheic Patients in Dessie Town, Northeast Ethiopia

PONE-D-20-27382R1

Dear Dr. Tsegaye,

We’re pleased to inform you that your manuscript has been judged scientifically suitable for publication and will be formally accepted for publication once it meets all outstanding technical requirements.

Kind regards,

Iddya Karunasagar

Academic Editor

PLOS ONE

Additional Editor Comments (optional):

All reviewer comments addressed satisfactorily

Reviewers' comments:

Reviewer's Responses to Questions

**Comments to the Author**

1. If the authors have adequately addressed your comments raised in a previous round of review and you feel that this manuscript is now acceptable for publication, you may indicate that here to bypass the “Comments to the Author” section, enter your conflict of interest statement in the “Confidential to Editor” section, and submit your "Accept" recommendation.

Reviewer #1: All comments have been addressed

2. Is the manuscript technically sound, and do the data support the conclusions?

Reviewer #1: Yes

3. Has the statistical analysis been performed appropriately and rigorously? 

Reviewer #1: Yes

4. Have the authors made all data underlying the findings in their manuscript fully available?

Reviewer #1: Yes

5. Is the manuscript presented in an intelligible fashion and written in standard English?

Reviewer #1: Yes

6. Review Comments to the Author

Reviewer #1: Thank you for the change you've made. I have no further comments, good luck with you continued work.

7. PLOS authors have the option to publish the peer review history of their article (what does this mean?). If published, this will include your full peer review and any attached files.

Reviewer #1: No

---

## [Editor Report · Acceptance letter]

4 Dec 2020

PONE-D-20-27382R1 

Prevalence of Enteric Pathogens, Intestinal Parasites and Resistance Profile of Bacterial Isolates among HIV Infected and Non-Infected Diarrheic Patients in Dessie Town, Northeast Ethiopia 

Dear Dr. Tsegaye:

I'm pleased to inform you that your manuscript has been deemed suitable for publication in PLOS ONE. Congratulations! Your manuscript is now with our production department. 

Kind regards, 

on behalf of

Dr. Iddya Karunasagar 

Academic Editor

PLOS ONE